# Different Approaches for Case-Mix Adjustment of Patient-Reported Outcomes to Compare Healthcare Providers—Methodological Results of a Systematic Review

**DOI:** 10.3390/cancers13163964

**Published:** 2021-08-05

**Authors:** Nora Tabea Sibert, Holger Pfaff, Clara Breidenbach, Simone Wesselmann, Christoph Kowalski

**Affiliations:** 1German Cancer Society, Kuno-Fischer-Str. 8, 14507 Berlin, Germany; breidenbach@krebsgesellschaft.de (C.B.); wesselmann@krebsgesellschaft.de (S.W.); kowalski@krebsgesellschaft.de (C.K.); 2Institute of Medical Sociology, Health Services Research, and Rehabilitation Science, Faculty of Medicine and University Hospital Cologne, Faculty of Human Sciences, University of Cologne, 50933 Cologne, Germany; holger.pfaff@uk-koeln.de

**Keywords:** patient-reported outcomes, case-mix adjustment, healthcare provider comparison, quality improvement

## Abstract

**Simple Summary:**

Patient-reported outcomes need to be reported with case-mix adjustment in order to allow fair comparison between healthcare providers. This systematic review identified different approaches to case-mix adjustment, with wide variation between the various approaches.

**Abstract:**

Patient-reported outcomes (PROs) are increasingly being used to compare the quality of outcomes between different healthcare providers (medical practices, hospitals, rehabilitation facilities). However, such comparisons can only be fair if differences in the case-mix between different types of provider are taken into account. This can be achieved with adequate statistical case-mix adjustment (CMA). To date, there is a lack of overview studies on current CMA methods for PROs. The aim of this study was to investigate which approaches are currently used to report and examine PROs for case-mix-adjusted comparison between providers. A systematic MEDLINE literature search was conducted (February 2021). The results were examined by two reviewers. Articles were included if they compared (a) different healthcare providers using (b) case-mix-adjusted (c) patient-reported outcomes (all AND conditions). From 640 hits obtained, 11 articles were included in the analysis. A wide variety of patient characteristics were used as adjustors, and baseline PRO scores and basic sociodemographic and clinical information were included in all models. Overall, the adjustment models used vary considerably. This evaluation is an initial attempt to systematically investigate different CMA approaches for PROs. As a standardized approach has not yet been established, we suggest creating a consensus-based methodological guideline for case-mix adjustment of PROs.

## 1. Introduction

There is a growing body of evidence that patient-reported outcomes (PROs) are effective in improving cancer patients’ health when used for patient monitoring or as a tool during patient consultations for better evaluation of patients’ symptoms or functional impairment [1,2,3,4]. Following landmark studies by Basch et al. [5,6] and Denis et al. [7], many approaches for incorporating (electronic) PROs into routine clinical cancer care have recently been developed—although some approaches also date back 15 years or more [8,9]. Due to cost and organizational barriers [10], only a small proportion of patients are currently benefiting from these approaches, but PROs are on the wish list of many service providers and healthcare systems and can be expected to find their way into more and more national cancer plans in the near future [11].

In addition to these approaches based on individual patients, PROs can also be used for another purpose in routine clinical care—quality assurance—in the sense of quality improvement initiatives. For this purpose, aggregated data at the level of healthcare providers are compared with those of other providers, resulting in rankings or benchmark reports. PROs are then regarded in a similar manner as other quality metrics: if suboptimal outcomes are observed at the provider level, the providers are typically asked to use these comparisons as part of their quality management in order to improve care. This function has been prominently advocated by, among others, the International Consortium on Health Outcomes Management (ICHOM), for whom PROs are a central part of the standard datasets used to allow comparisons. Building on the ICHOM standard dataset for localized prostate cancer [12], for example, the multinational TrueNorth Global Registry has been set up [13], with over 200 participating sites in 15 countries and with national substudies, e.g., [14]. It has been argued in the past that these two functions (with approaches at the individual patient level and at the provider level) can work hand in hand, but examples of this are as yet scarce [15].

When PROs are used as quality metrics to compare healthcare providers, special attention needs to be given to the different case-mixes present in order to allow fair reporting [16]. If PROs were to be reported without adjustment, services with less severely affected patients or otherwise favorable patient characteristics, such as fewer comorbidities or earlier disease stages, might appear to be better than they actually are, and vice versa. This applies to quality metrics in general and may lead to inappropriate incentives such as refusal to treat specific patients—especially if the providers are ranked and the results are publicly reported, with the providers’ names not being concealed. The practice of refusing patients in order to avoid costs or unfavorable outcomes in competitive healthcare markets is often referred to as “cream skimming” and is highly applicable to PRO reporting. According to Iezzoni’s framework of case-mix adjustment (CMA), a healthcare outcome can be described as a function of “intrinsic patient-related risk factors, treatment effectiveness, quality of care” and “random chance” [16]. Thus, when PROs are used to compare providers, effects summarized as “intrinsic patient-related risk factors” must not distort the comparison results. CMA (which is sometimes more generally referred to as “risk adjustment”) aims to account for such differences in providers’ case-mixes using a statistical procedure. The goal of CMA is to allow fair and reliable comparison of providers relative to the given outcome(s). Its aim is to allow “like-to-like comparisons,” as described by Iezzoni in her introductory chapter describing the purpose of CMA [16]. Ultimately, CMA is essential for the credibility and usefulness of provider comparisons.

Different approaches to CMA have been used in the past for quality metrics other than those based on PROs, and differences in approaches may lead to substantial differences in the reported outcomes and in the ranking of providers and services. There have been few publications on how to adjust the case-mix for PROs, and there is a lack of systematic evaluation of published and proposed CMA approaches. The aim of the present systematic review was to identify methods used for CMA in provider comparisons using PROs, with a particular focus on the adjustors included in the CMA models and the statistical approaches used. Recommendations on the choice of future CMA methodology—as suggestions for a possible CMA guideline—are also derived from this.

## 2. Methods

### 2.1. Studies and Methodological Decisions of Interest

This systematic review adapted the participants, interventions, comparisons, outcomes, study designs (PICOS) template, with omission of the intervention (I) and comparator (C) categories, which were not applicable here. Any patients treated by various healthcare providers were therefore examined as participants (P), PROs as outcomes (O), and observational studies and secondary analyses of interventional studies were included as study designs (S).

Iezzoni [16] proposed four main questions that can be used for guidance in finding the “right” CMA approach:Risk of what outcome?Over what time frame?For what population?For what purpose?

These questions (referred to as “Iezzoni’s four”) are therefore used in this review to evaluate the different CMA approaches together with the adjustors used, the criteria with which the adjustors were chosen, the statistical model used, and the way in which the results were reported.

### 2.2. Eligibility Criteria and Information Sources

This review includes observational studies or secondary analyses of interventional studies published between January 1992 and January 2021 written in English, French, or German, in which PROs are used to compare the quality of care between recognizable healthcare providers and in which any kind of case-mix adjustment of PROs is used. PROs were considered to be “instruments to elicit information from patients on self-reported health status [and] outcomes” in accordance with the definition provided by the Organization for Economic Cooperation and Development (OECD) [17]. Studies were included if they used at least one adjustor that was not the PRO baseline score. Exclusion criteria were: studies that did not use (case-mix-adjusted) PROs for provider comparison, or had no provider comparison or no CMA or did not explain their CMA; those that used patient-reported experience measures (including satisfaction surveys) as opposed to the above definition of PROs (patient-reported *experience* measurements measure—in contrast to PROs—“how patients experience health care and refer to practical aspects of care, such as accessibility, care coordination and provider-patient communication” [17] and are as such not healthcare outcomes). The electronic database MEDLINE was used as the information source. Citations and references in the studies included were also screened for eligibility. The review was registered at PROSPERO (CRD42020180420).

### 2.3. Search and Study Selection

The search string was developed iteratively, with test searches being conducted in the MEDLINE database by N.T.S. and C.K. The study group developed the final search strategy with support from an experienced systematic reviewer, combining independent and medical subject headings (MeSH) terms in accordance with the PRESS guideline statement [18]. The final search string contained variations of the terms: “humans” (P), “patient-reported outcomes” or “patient-reported outcome measures” (O), and “risk adjustment” or “case-mix adjustment.” The full search string can be found in Appendix A.

C.K. and N.T.S. first independently screened all titles and abstracts. N.T.S. searched for available full texts, which were then independently reviewed by C.K. and N.T.S. relative to the inclusion criteria. Any divergent results were discussed by C.K. and N.T.S. to obtain a consensus.

### 2.4. Data Collection Process, Data Items, and Risk of Bias in Individual Studies

C.K. and N.T.S. independently used a standardized data extraction form that was developed deductively using Iezzoni’s earlier research on risk adjustment [16]. Data items included in the extraction form were: type of provider, Iezzoni’s four, information on adjustors (which ones, and how they were chosen), CMA methodology, and reporting of case-mix-adjusted PROs. In addition, quality assessment was carried out using the Critical Appraisal Skills Programme (CASP) checklist for cohort studies [19].

### 2.5. Summary Measures and Synthesis of Results

Data synthesis was carried out by qualitatively summarizing the extracted information on statistical methods used for CMA, adjustors, and reporting of provider comparisons. Since there was wide heterogeneity among the CMA approaches and PROs applied, it was not possible to pool the findings quantitatively as a meta-analysis. However, where possible, similar case-mix adjustment approaches are presented in bundled form. No additional subgroup analysis was carried out in any of the studies.

### 2.6. Risk of Bias across Studies

As the purpose of this methodological review was to provide an overview of the statistical methods used for adjustment, and the studies included were conducted in different populations, over different time periods, and with different PROs, it was not possible to carry out a meta-analysis. Evaluation of the risk of bias among the studies included was therefore not applicable.

## 3. Results

### 3.1. Study Selection and Characteristics

A total of 640 studies were identified in MEDLINE using the search string. After screening of the titles/abstracts and full text, nine studies were included for the qualitative synthesis (Figure 1: PRISMA flowchart). Two studies were added after the references from these nine studies had been screened. Thus, a total of eleven studies were included in this analysis. An overview on study characteristics can be found in Table 1.

### 3.2. Quality of Included Studies (CASP Evaluation)

All studies had a clear focus, precise and mostly applicable results, and fitted with other evidence. Follow-up information was partly insufficient or could not be examined. Details are provided in the CASP results Table A1 in Appendix B.

### 3.3. Provider Settings

Five inpatient settings [20,21,22,23,24], five outpatient settings [25,26,27,28,29], one mental health services setting [26], and three rehabilitation settings [20,28,29] were identified. Ten studies included patients with orthopedic conditions [20,21,22,23,24,25,27,28,29,30]. Three of the studies included were set in National Health Service-funded hospitals in the United Kingdom [21,23,30]. Three studies were set in hospitals in the United States, taking part in the Focus on Therapeutic Outcomes (FOTO) initiative [25,28,29].

### 3.4. “Iezzoni’s Four”

This synthesis follows Iezzoni’s approach on analyzing risk adjustment. Iezzoni’s four main questions are therefore covered in the following sections [16]. Detailed information is provided in Table 2.

#### 3.4.1. Risk of What Outcome?

The studies included a total of 17 different PRO instruments: seven generic PRO questionnaires (PCS [31], PF-10 [32], SF-12 [31], ADL [33], Lehman Quality of Life Interviews [34], EQ-5D [35], including one rehabilitation-specific one: IRES [36]) and ten condition-specific instruments (VAS-BP, VAS-LP [37], OHS, OKS [38], MDQ [39], NDI [40], ODI [41], MacNew [42], SMFA [43], and LCAT [44]).

#### 3.4.2. For What Population?

All of the studies included explored PROs for an adult population. Ten studies were conducted in orthopedic settings [20,21,22,23,24,25,27,28,29,30], and one investigated patients with psychiatric conditions [26]. None of the studies that were identified investigated PROs in populations of cancer patients.

#### 3.4.3. Over What Time Period?

Most of the studies examined the development of PRO scores from baseline (e.g., at admission, before the start of therapy) and a predefined or treatment success-defined post-therapeutic time point. In some studies, the time period of PRO data collection was not reported or specified [24,27,28]—for example, if the post-therapeutic end point was measured at the end of an intervention, of which the duration could vary, e.g., [28].

#### 3.4.4. For What Purpose?

The main purpose of the studies included was to obtain inter-provider comparisons to measure performance, in some cases resulting in a (clinical) ranking [25] or in the identification of special expertise [28].

### 3.5. Selection of Adjustors

There are several approaches for choosing adjustors for CMA. The studies included only partly reported on the selection process. Two studies stated that they chose the adjustors that were used on the basis of statistical parameters: Resnik and Hart used bivariate analysis to identify suitable adjustors [28], and Deutscher et al. employed a back-step regression model [25]. Farin et al. and Khor et al. based their choice of adjustors on earlier findings by their research groups [20,22]. Table 2 provides further information.

### 3.6. Patient Characteristics Included in the CMA Models

All of the studies included sociodemographic characteristics (such as sex, age) of their patients and baseline PRO scores as adjustors in the CMA models used. All of the studies chose comorbidities and/or a measure of the patient’s general physical status (i.e., specific comorbidities, number of comorbidities, Charleston Comorbidity Index, American Society of Anesthesiologists score, etc.) as adjustors. Most of the studies that were included featured disease-specific information or information about treatments previously received by the patients.

### 3.7. Provider Characteristics

Lutz et al. included provider characteristics (the American state in which the study was located) as an adjustor and Sivaganesan et al. included information about surgeons (years in practice, fellowship training) [24,27].

### 3.8. Statistical Approaches

The statistical approaches used differed in relation to the intended presentation of the adjusted results. Some studies calculate scores that keep the scale of the initial PRO score, while others keep the scale but present “performance indicators”—typically the difference between the observed score and the score expected on the basis of the patient case-mix. These approaches are referred to in this article as “unit/scale-preserving approaches.” Other studies compute ratios of the observed scores divided by the expected scores, or only calculate expected scores. These approaches thus do not present results that are consistent with the units of the original PRO score and are referred to in this article as “unit/scale-aberrant approaches.”

A common differentiation made in the statistical models used for CMA is that between *direct* and *indirect* standardization [45]. Briefly, direct approaches use a matched subsample from each provider to compare the outcomes for those patients with those in a standardized population, whereas indirect approaches use the whole sample from the providers and adjust the outcomes using a modeling procedure, typically regression analyses. All of the studies included in this review used indirect standardization methods. However, they differ in the way in which the adjusted scores are calculated and presented. These differences are therefore examined in greater depth in the following paragraphs.

Since both the topic itself and differences in CMA methods are quite complex, the different statistical approaches are illustrated by using an example based on the Prostate Cancer Outcomes (PCO) Study data [14] (*described in italics*):


*To compare outcomes of prostate cancer centers one year after treatment—e.g., surgery—certified prostate cancer centers take part in a survey collecting information on EPIC-26 scores. The EPIC-26 is a prostate cancer-specific PRO questionnaire resulting in five scores, among them an incontinence score—with 0 being the lowest and 100 the highest possible score [46]. For our example, we would like to compare Center A, Center B, and Center C using the following metrics:*



*Unadjusted—i.e., observed—mean incontinence scores (1 year after surgery): 26 (Center A), 37 (Center B), 31 (Center C), 30 (all patients in all centers)*



*Expected mean incontinence scores based on linear regression models (using the following adjustors: baseline EPIC-26 scores, age, tumor stage, etc.): 20 (Center A), 33 (Center B), 40 (Center C).*


#### 3.8.1. Calculation of Transformed Adjusted Scores (Unit/Scale-Preserving Approaches)

Two studies used the observed and previously calculated expected values to transform them into adjusted scores that can be interpreted using the initial PRO score’s unit [23,28]: Nuttal et al. firstly calculated the ratio $\frac{observed}{expected}$ for every provider and multiplied those ratios with the overall population’s observed average [23]:


(1)$$adjusted\;score=\frac{observed}{expected}\;*\;overall\;observed\;average.$$



*In our example, the adjusted scores were:*



*Center A:*
$\frac{26}{20}\;*\;30\;\approx \;39$



*Center B:*
$\frac{37}{33}\;*\;30\;\approx \;33$



*Center C:*
$\frac{30}{40}\;*\;30\;\approx \;26$



*These scores are all within the original EPIC-26 limits (0–100) and are interpreted identically. The adjusted scores provide information about what the incontinence score of an “average” prostate cancer patient would be in the respective center.*


Resnik and Hart calculated the differences between observed and expected values [28]. The resulting adjusted (or “performance”) scores were not within the limits of the original PRO scale (e.g., for a positive PRO scale, the differences might become negative), but they are interpretable using the original units:(2)adjusted score=expected−observed

Deutscher et al. and Farin et al. used the standardized residuals of the final case-mix regression models as “adjusted outcomes” [20,25]. This approach is therefore very similar to the adjustment method used by Resnik and Hart (residuals are defined as the difference between the observed and expected (“fitted”) values in regression analyses).


*Using the Resnik and Hart approach, we obtain the following adjusted scores using our example:*



*Center A: +6*



*Center B: +4*



*Center C: −10*



*These adjusted scores provide information about how the centers perform in relation to their expected outcomes (e.g., Center C is performing less well than expected, whereas Center A is achieving better results) using the same unit as in EPIC-26. It would, for example, be possible to use the EPIC-26 minimally important differences (MIDs; lower threshold for incontinence: 6 [47]) to interpret these results better: in Center B, the “better performance” would be less than one MID, and therefore not perceptible for the patients, whereas a patient treated in Center C would notice the poorer performance.*


Varagunam et al. used a similar approach to Nuttal et al., but based on the residuals, firstly, they calculated the difference in the observed and the expected value (ergo, the residuals) for every provider and then added this to the overall population’s observed average [30]:(3)$$adjusted\;score=\left(expected-observed\right)+overall\;observed\;average.$$


*In our example, the adjusted scores were:*



*Center A:*
$\left(26-20\right)+30=36$



*Center B:*
$\left(37-33\right)+30=34$



*Center C:*
$\left(40-30\right)+30=20$



*These scores are all within the original EPIC-26 limits (0–100) and are interpreted identically. The adjusted scores provide information about what the incontinence score of an “average” prostate cancer patient would be in the respective center.*


#### 3.8.2. Calculation of Transformed Adjusted Scores (Unit/Scale-Aberrant Approaches)

Two studies calculated the ratio $\frac{observed}{expcted}$ and used this “ratio indicator” as an adjusted outcome [26,27].


*The ratio indicators for the PCO examples are:*



*Center A:*
$\frac{26}{20}\;\approx \;1.3$



*Center B:*
$\frac{37}{33}\;\approx \;1.12$



*Center C:*
$\frac{30}{40}\;\approx \;0.75$



*These adjusted scores are not interpretable using the original EPIC-26 unit. However, they quantify the deviation from the expected outcome: Centers A and B are both performing better than expected, resulting in a ratio > 1, whereas Center C is performing less well than expected, with a ratio < 1.*


One study investigated the proportion of patients who experienced an improvement of at least one minimally important difference (MID) as the outcome of interest—i.e., they dichotomized the original PRO score [22]. They then calculated the expected number of events per provider on the basis of a previously published risk calculator, calculated the observed/expected ratio, and finally calculated the case-mix-adjusted event rate by multiplying the O/E ratio by the overall state-wide average.


*The MID for the EPIC-26 incontinence score is 6, as described above. Let us assume that 20% of the patients of Center A, 40% of the patients of Center B, and 30% of the patients of Center C had an improvement of at least 6 points concerning their incontinence score after surgery. Appling the risk calculator for the centers’ populations, for Center A, 18% would have been expected, for Center B 35%, and for Center C 50%. For the state-wide population, an improvement of 30% is already known (this is a theoretical example). Using the “MID approach,” the adjusted scores would be:*



*Center A:*
$\frac{20}{18}\;*\;30\;\%\;\approx \;27\;\%$



*Center B:*
$\frac{40}{35}\;*\;30\;\%\;\approx \;34\;\%$



*Center C:*
$\frac{30}{50}\;*\;30\;\%=18\;\%$


#### 3.8.3. Other

Sivaganesan et al. calculated the expected values for the providers examined, on the basis of their case-mix. However, the authors did not seem to use the expected values to compute any further “transformed” adjusted scores. They used a multilevel model [24].


*For our example, the “adjusted scores” would be the same as the expected ones:*



*Center A: 20*



*Center B: 33*



*Center C: 40*



*Center C, which performed 10 points less well than expected, would be top ranked following this approach.*


Gutacker et al. used another approach: based on a multilevel model using an hierarchical Bayesian approach with the PRO score (EQ-5D) used as an ordinal scaled variable, the predicted probabilities for each outcome per provider were calculated and reported as a quality metric [21].

Using a multilevel model as well, Gozalo et al. reported a provider ranking based on the predicted providers’ random intercepts [29]. 

#### 3.8.4. Minimum Case Number Requirements for Statistical Modeling/Adjustment

Seven studies did not report any minimum case number requirements at the provider level for adjustment. The other studies differed widely in the minimum numbers of patients per provider that were required for adjustment. Gozalo et al. and Resnik and Hart set a minimum of eight patients per provider [28,29], whereas Farin et al. required at least 200 patients per provider [20]. Deutscher et al. only included providers in which at least 50% of the patient records were complete [25]. Most of the studies did not provide any explanations for the thresholds chosen.

## 4. Discussion

The results of the present systematic literature review on how to case-mix adjust PROs for provider comparison show that there is substantial variation in the literature with regard to several methodological criteria.

Firstly, the review shows that a uniform approach to case-mix adjustment of patient-reported outcomes does not currently exist. In most cases, regression models were used to adjust for relevant patient characteristics that differed between providers, with expected outcomes being calculated. Five of the eleven studies that were included made comparisons between expected and observed values, and three of these studies ultimately calculated an outcome that was evaluable in terms of the scale of the original one. One study seemed to only calculate fitted values and based the resulting provider ranking on them, which seems to be a questionable adjustment approach since providers with a “favorable” case-mix (e.g., younger or healthier patients) will always perform better than providers with a “difficult” case-mix if using this CMA approach. This can result in a biased ranking and does not provide information on the actual quality of care.

Secondly, there were also considerable differences between the studies included in the way in which expected values were calculated, and thus in the selection of adjustors. Relevant sociodemographic information and important clinical parameters (e.g., comorbidities) were included in most studies, and baseline patient-reported outcomes were included in all of them. There is little debate on including baseline PROs, as it is known from many observational and validation studies that PROs can vary considerably between care providers before the start of treatment even when adjustment is made for other patient characteristics [49]. It is thus essential to take baseline PROs into account in the adjustment.

Thirdly, nine of the studies included only reported patient characteristics, while two also included the characteristics of the providers that were being compared, such as information about the training of surgeons. Not taking provider characteristics into consideration is consistent with the purpose of CMA, since if outcomes are a function of patient characteristics, treatment effectiveness, and quality of care, and if the aim of the survey is to compare treatment quality, then considering provider characteristics may lead to over-adjustment. However, there are situations in which it might be justifiable to take provider characteristics into account—for example, if some contextual factors (such as state legislation) affect the outcomes that are exogenous, beyond the control of the provider. In principle, it is preferable to investigate provider characteristics in a further step in order to explain any differences in treatment quality in greater detail. This was done, for example, by Resnik and Hart [28], who looked at associations between provider characteristics and whether a practitioner was classified as an “expert physical therapist.” They found no association between years of experience and this expert status, for example.

Surprisingly, the MEDLINE search did not identify any cancer-related studies. However, in addition to the peer-reviewed studies that were included, there are also public quality initiatives designed to inform the general population about the quality of care in cancer settings, some of which include PROs. For example, in the National Prostate Cancer Audit (NPCA) in the United Kingdom, post-therapy PROs (EPIC-26) are adjusted for case-mix and reported using funnel plots [50]. This initiative, however, does not take baseline PROs into account for the adjustment, and the results have not yet been reported in peer-reviewed journals. Moreover, we excluded five studies that did use case-mix-adjusted PROs to compare providers, but unfortunately did not explain their adjustment methods [51,52,53,54,55]. However, those studies highlight that PROs—when case-mix adjusted—are valid and valuable parameters to compare different providers. 

### Limitations

We are aware of several limitations of our study: to begin with, only the MEDLINE database was used for the literature search. To find any other peer-reviewed research, all references of eligible studies were therefore scanned, and two more articles were included in this systematic review.

Moreover, articles not published in English, German, or French were excluded (language bias), which may result in a selection bias. Although a search strategy strictly following the PRISMA recommendations was used and an additional reference screening was performed, it is possible that some relevant studies were missed if the specific keywords were not used by either the authors or the MEDLINE database.

Since the included studies showed a huge variety in terms of the statistical methods used, it was not possible to perform a meta-analysis. However, as those statistical approaches themselves were the research subject, this was not expected.

For assessing the quality and risk of bias of the included studies, the CASP tool results are presented. The included studies are all retrospective analyses of prospectively collected data and the appraisal tools for cohort studies were mostly applicable. However, most of those appraisal tools incorporate items that are not applicable for our purposes, such as the assessment of the exposure of the outcome (e.g., [56]) and we decided to only use the CASP tool, which does not provide a numerical scale for assessing the quality of the studies, and thus the rating of each study is subjectively dependent on the reader’s main focus. This subjectivity may be a limitation, however, the use of the CASP tool results in a greater flexibility and usability of the risk of bias evaluation. 

The review was conducted strictly according to PRISMA. We employed a concise and literature-based search string that was critically reviewed by multiple peers. The methodology used for this review was assessed for validity by the PROSPERO Foundation, where a protocol was pre-registered. A thorough quality assessment focusing on methodological issues was performed for each included study.

## 5. Conclusions and Recommendations

All in all, the variety of the approaches and the small numbers of published articles on the topic underscore the uncertainties concerning the best way of carrying out case-mix adjustment of PROs. To report interpretable PROs for all stakeholders (patients, clinicians, decision makers, researchers, etc.), it would be desirable for a more consistent methodology to be used across approaches.

Moreover, taking the heterogeneity of our findings into account, we would like to point out two additional major challenges of CMA of PROs—especially in contrast to non-PRO quality metrics—for which we see a need for further research:

Firstly, while most other quality metrics are rate based, with only two possible response categories (such as dead/alive or complication/no complication), PROs typically come in the form of continuous measures. PROs thus require data collection before and after an intervention (such as a hospital stay), because a patient’s functional status can change gradually, whereas in the other example, he/she is necessarily alive and could not have had an intervention-related complication before an intervention. Continuous measures with two points of measurement, however, require a number of analytical decisions associated with statistical pitfalls. For example: should *changes* from before and after an intervention be reported, or should adjusted *scores after* an intervention be presented that include the pre-intervention score as an adjustor? The included studies in this review mostly adjusted the scores after intervention. However, some study groups decided to report changes (e. g., [52]) and qualitative research on how to present PROs to clinicians often highlights the importance of presenting trends over time [56,57].

Secondly, while traditional quality metrics data are typically collected without—or with little—active contribution from the patient and are thus mostly documented for all patients (although with varying documentation quality), PRO data collection requires active involvement of the patients, who have to complete the questionnaires. This leads to patient selection and may create substantial bias. For example, patients with some characteristics are less likely to complete questionnaires (due to factors such as age, educational level, or their degree of trust in the healthcare system) [58]. Similarly, providers may be reluctant to include some patients because they expect them to have a negative impact on their results—e.g., patients with more comorbidities tend to have poorer PRO scores [59]. Correcting such potential biases is difficult because of the unknown characteristics of the unsurveyed patients, so weighting is scarcely possible. In practice, this may lead to a decision to include only those providers in comparisons for which a minimum number (or proportion) of patients are included in the analytical dataset. Thus, in our opinion, quality improvement initiatives based on PRO comparisons should be encouraged to include all eligible patients and report on potential selection biases.

Due to the lack of comparisons between different adjustment methods for the same datasets, there is no strong scientific evidence for favoring one approach over the others. Recommendations on what a “gold standard” should include are therefore difficult to make at present. We would therefore cautiously derive three recommendations as a starting point for further discussion, on the basis of what appear to be successful practices in the studies included in this review. In order to keep the threshold for comparisons between treatment outcomes with the help of PROs as low as possible, so that they are accessible to a broad public, we would recommend the following on the basis of the present findings:

Firstly, with regard to baseline PROs, it appears to be common methodological practice to include them. All of the studies feature them either as an adjustor in the model, or as part of the dependent variable when the difference between baseline and follow-up PRO is used. The reasons for doing so are well elaborated on in some of the studies included [23,27].

Secondly, a number of the studies emphasized that the CMA should be accessible by different stakeholders (e.g., clinicians, patients) and should therefore be easily interpretable and offer additional information that could be used for better selection, or to identify providers with good performance [20,23,25]. In our view, this notion favors unit/scale-preserving approaches, so that a new “unit” does not need to be learned.

Thirdly, in line with most of the studies included, provider characteristics should not be included unless there are strong reasons for this such as contextual factors that are beyond the providers’ control.

In the absence of strong evidence favoring one approach over any of the others, we would suggest that further research should be carried out, including comparison of approaches both in terms of the statistics used and also the comprehensibility of the reported results for the target audiences of the provider comparisons. Among the many open questions are minimum case thresholds, whether to take the hierarchical structure of the data into account using multilevel models, as well as measuring the impact that CMA has on rankings and the threshold at which case-mix-adjusted differences in PROs should be considered relevant at all. The variation in CMA approaches underlines the importance of establishing a methodological guideline or “checklist” that could be used, for example, for quality improvement initiatives aiming to use PROs for provider comparisons. As the evidence is still scarce, we would recommend—alongside further research on the benefits and pitfalls of different CMA methods—a consensus-based approach to the development of such a guideline.

## Figures and Tables

**Figure 1 cancers-13-03964-f001:**
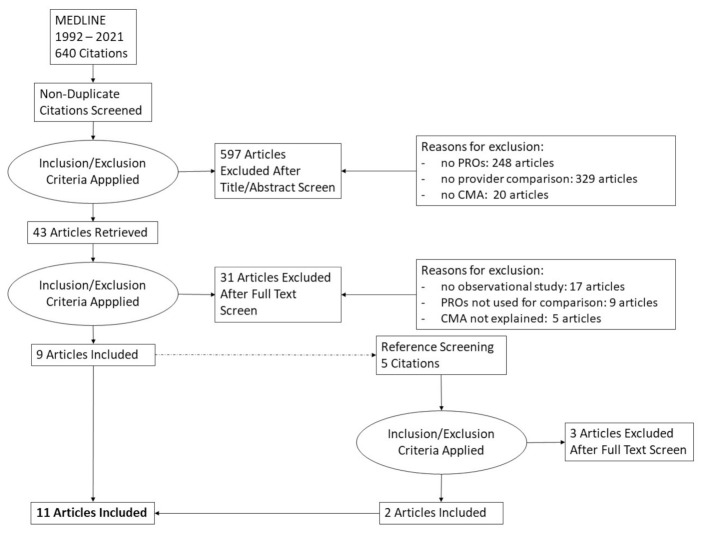
PRISMA flowchart. PRO, patient-reported outcome; CMA, case-mix adjustment.

**Table 1 cancers-13-03964-t001:** Characteristics of eligible studies.

Author	Study Design	Participants/Setting	Data Sources	Study Size (Complete/Included Datasets)	Outcome (PRO)	No. of Variables Used for Case-Mix Adjustment (and Specific Adjustors)	Model R^2^
Deutscher et al., 2018 [25]	Retrospective analysis of prospectively collected data	Adult patients with lower back pain treated in outpatient physical therapy clinics (2014–2016)	FOTO	*n* = 341,642 patients *n* = 6934 clinicians *n* = 2107 hospitals	LCAT	11: LCAT score at admission, age, sex, acuity, payer, surgical history, physical exercise history, medication/previous treatment, type of surgery, comorbidities	0.373
Farin et al., 2009 [20]	Analysis of quality assurance data	Quality assurance report for statutory rehabilitation institutions in Germany on: 1, musculoskeletal; 2, cardiological; and 3, neurological diseases	QS Reha^®^	1: *n* = 4045 patients in *n* = 27 hospitals 2: *n* = 2503 patients in *n* = 25 hospitals 3: *n* = 1477 patients in *n* = 12 hospitals	1: IRES 2: SMFA 3: MacNew	Reported elsewhere [48]: 10: age, sex, somatic, functional and psychosocial status at admission, diagnosis, multimorbidity, treatment motivation, application for pension, education	Not reported
Gozalo et al., [29]	Retrospective analysis of prospectively collected data	Outpatient rehabilitation clinics for patients with lower back pain	FOTO	*n* = 90,392 patients *n* = 2040 therapists *n* = 538 clinics	FOTO overall health status measure: OHS, PCS, PF-10	8: PRO at intake, age, gender, onset, surgery count, functional comorbidity index, fear avoidance level, payer type	OHS: 0.416 PCS: 0.345 PF-10: 0.421
Gutacker et al., 2013 [21]	Retrospective analysis of prospectively collected data	Analysis of hospital performance variation in hip replacement surgery	NHS PROMs and routinely collected inpatient data (HES)	*n* = 21,565 patients *n* = 153 hospitals	EQ-5D	10: pre-treatment PROs, age, gender, deprivation index, weighted Charlson Index of comorbidities, number of additional comorbidities (not included in Charlson Index), time between pre-operative assessment and admission, primary surgery, revision surgery, treatment	Not reported
Hendryx et al., 1999 [26]	Retrospective analysis of prospectively collected data	Development of outcome risk adjustment models for public mental health outpatient treatment programs	Patient-reported, case manager ratings, management information system data	*n* = 289 patients *n* = 6 mental health agencies	SF-12, ADL, Lehman Quality of Life Interviews	8: sex, age, race, presence of severe primary diagnosis (major depression, schizophrenia, bipolar disorder), baseline levels of substance abuse, baseline PRO, baseline quality of life, baseline satisfaction with services	0.34
Khor et al., 2020 [22]	Retrospective analysis of prospectively collected data	Analysis of variation in PROs 1 year after elective lumbar fusion surgery across surgeons and hospitals	Spine Care and Outcomes Assessment Program (Washington State, USA)	*n* = 737 patients *n* = 17 hospitals *n* = 58 surgeons	ODI	11: age, sex, insurance status, race, ASA, smoking status, prior spine surgery, diagnosis, opiate use, asthma, baseline PRO	Not reported
Lutz et al., 2020 [27]	Retrospective analysis of prospectively collected data	Benchmarking physical therapist study	ATI Patient Outcomes Registry	*n* = 182,276 patients *n* = 2799 physical therapists	MDQ, NDI	10: sex, age, BMI, initial PRO, payer type, physical component score and mental component score of the Veterans RAND 12-Item Health Survey, state of physical therapy services, type of pain (acute/chronic), comorbidities	MDQ: 0.19 NDI: 0.19
Nuttall et al., 2015 [23]	Retrospective analysis of prospectively collected data	Feasibility study to analyze case-mix adjustment methodology for elective surgery	NHS PROMs and HES	*n* = 30,555 patients *n* = 237 providers	OKS	11: baseline PRO, sex, ethnicity, age, deprivation index, assistance for completing questionnaires (baseline and follow-up), disability, previous surgery, comorbidities, type of surgery, length of post-operative stay	0.258 (ordinary least square model) 0.257 (fixed effects model)
Resnik & Hart 2003 [28]	Retrospective analysis of prospectively collected data	Outpatient rehabilitation clinics for patients with lower back pain	FOTO	*n* = 24,276 patients *n* = 930 therapists *n* = 354 hospitals	FOTO overall health status measure (OHS), PCS, PF--10	8: age, employment, exercise history, sex, intake PRO, onset, reimbursement, surgery	OHS: 0.416 PCS: 0.345 PF-10: 0.421
Sivaganesan et al., 2018 [24]	Retrospective analysis of prospectively collected data	Report of a risk-adjusted ranking of spine surgeons and sites performing elective lumbar surgery	Quality and Outcome Database (QOD)	*n* = 8834 patients *n* = 124 surgeons *n* = 21 sites	ODI, EQ-5D, VAS-BP/VAS-LP	22 (19 at patient level: age, BMI, ethnicity, education, smoking status, opioid use, comorbidities, pre-operative symptoms, motor deficit, ASA, symptom duration, interbody graft placement, worker’s compensation, liability claims, insurance status, employment, baseline PRO); 3 constructs at surgeon level: site ID, years in practice, fellowship training)	Not applicable (Bayesian model)
Varagunam et al., 2015 [30]	Retrospective analysis of prospectively collected data	Feasibility study analyzing different approaches on how to adjust PROM scores for individual consultant comparison for elective surgery	NHS PROMs	N = 65,465 (hip), 68,107 (knee) and 38,965 (hernia) patients N = 948 (hip), 1130 (knee) and 974 (hernia) consultants N = 183 (hip), 188 (knee) and 197 (hernia) clinics	OHS, OKS, EQ-5D	6: age, sex, deprivation index, comorbidities, previous surgery, baseline PRO	Not reported

ADL, Activities of Daily Living; ASA, American Society of Anesthesiologist score, ATI, Athletic Therapeutic Institute; BMI, Body Mass Index; EQ-5D, European Quality of Life 5 Dimensions; FOTO, Focus on Therapeutic Outcomes; HES, Hospital Episode Statistics; IRES, *Indikatoren des Reha-Status* (Indicators for Rehabilitation Status); LCAT, Lumbar Computerized Adaptive Test; MDQ, Modified Low Back Pain Disability Questionnaire; NDI, Neck Disability Index; NHS, National Health Service (UK); ODI, Oswestry Disability Index; OHS, Oxford Hip Score; OKS, Oxford Knee Score; PCS, Physical Component Summary; PF-10, Physical Functioning Scale; PRO, Patient-Reported Outcome; PROM, Patient-Reported Outcome Measure; SF-12, Short Form 12; SMFA, Short Musculoskeletal Function Assessment questionnaire; VAS-BP/VAS-LP, Visual Analog Scale for Back Pain/for Leg Pain.

**Table 2 cancers-13-03964-t002:** Synthesis of case-mix adjustment methods.

Author	What Type of Provider?	Iezzoni’s Four				Which Adjustors?				Are Criteria for the Selection of Adjustors Met? ^1^		How is Case-Mix Adjustment Statistically Performed?	How are Case-Mix Adjustment Results Reported?
		Risk of What Outcome?	For What Population?	Over What Time Period?	For What Purpose?	Socio-Economic Information?	Baseline PROM Included?	Comor-Bidities?	Adjustment for ProVider Characteristics?	Are the Adjustors Associated with the Outcome?	Do Adjustors Vary Across Providers?		
Deutscher et al., 2018 [25]	Outpatient physical therapist clinics (US)	LCAT	Lower back pain patients treated in outpatient setting	Time between admission and discharge	Clinical ranking for performance measuring	x	x	x	–	Yes, assessed via back-step regression model approach	Not analyzed	Multiple regression, >50% complete patients per hospital and ≥10 complete cases per clinician per year required	Comparison between adjusted and unadjusted scores (agreement: 30%), ranking changed 70% of the hospitals, reported as percentile ranking
Farin et al., 2009 [20]	In- and outpatient rehab. institutions (Germany)	IRES, SMFA, MacNew	Patients in statutory rehab. institutions	Time between start and end of rehab.	Quality assurance	x	x	x	–	Not reported	Not reported	Multilevel regression, taking the standardized residuals in the prediction as the case-mix-adjusted outcome, >199 patients per institution required	95% confidence intervals reported graphically
Gozalo et al., 2016 [29]	Outpatient rehabilitation clinics (US)	FOTO CATs, either generic or body impairment specific	Patients treated in outpatient rehabilitation clinics (for orthopedic conditions)	At intake and discharge from outpatient rehabilitation	Identify expert physical therapists	x	x	x	-	Not assessed	Not reported	Generalized linear model with >7 patients per clinic required	Ranking of clinics based on predicted clinic random intercept (including 95% confidence interval)
Gutacker et al., 2013 [21]	NHS-funded hospitals (UK)	EQ-5D	Patients receiving hip replacement in an NHS-funded hospital	Time between before/day of admission and 6 months after surgery	Inter-provider comparison for performance measuring	x	x	x	–	Yes, assessed via regression modeling	Not reported	Multilevel linear regression and probit model, no minimum number of patients per hospital required	95% confidence intervals reported graphically
Hendryx et al., 1999 [26]	Outpatient mental health services (US)	SF-12 (functional status domain), Lehman Quality of Life Interview (Quality of Life domain)	Adult patients treated in outpatient mental health agencies	At baseline, 5 and 10 months	Inter-provider comparison for performance measuring	x	x	x	–	Not reported	Not assessed	Multiple linear regression, then calculating the ratio: observed/expected, no minimum number of patients per agency required	Observed score ranks and observed/expected ratio ranks are compared
Khor et al., 2020 [22]	In-patient hospitals performing elective spine surgery (USA)	ODI (probability of improvement beyond MID and reaching minimal disability level)	Adults treated with elective lumbar fusion surgery	At 0–60 days before surgery and at 12-month follow-up	Inter-provider comparison for performance measuring	x	x	x	–	Yes (association based on findings from earlier studies by the research group)	For all adjustors except for age, sex, and smoking status	1. Multiple logistic model 2. Calculation of an expected number of events per provider based on a previously published risk calculator 3. Calculation of observed/expected ratio 4. Calculation of a case-mix-adjusted event rate by multiplying the O/E ratio by the overall state-wide average; >9 patients per provider required	Adjusted and unadjusted scores, ICCs to describe the proportion of total variability accounted for by between-provider variance
Lutz et al., 2020 [27]	Outpatient physical therapists (USA)	MDQ, NDI	Adults with episodes related to the lower back and neck with no history of related surgery	Not specified	Benchmarking physical therapists	x	x	x	x (state of physical therapy services)	Not reported	Not assessed	1. Multiple linear regression 2. Calculation of an observed/expected ratio; >39 patients per physical therapist required	Reporting of counts of “outperforming, meeting, and underperforming” physical therapists (adjusted and unadjusted) based on 95% confidence intervals
Nuttall et al., 2015 [23]	NHS-funded hospitals (UK)	OKS	Patients treated by NHS providers for unilateral knee replacement	Before and 3 or 6 months after surgery	Inter-provider comparison for performance measuring	x	x	x	–	Yes (association based on findings from earlier studies)	Not analyzed	Multilevel linear regression: generalized least squares with fixed effects, general least squares with random effects, ordinary least squares, then calculating: (observed/expected) * observed mean overall; no minimum number of patients per hospital required	Differences between different adjustment scores reported, ranking reported as funnel plots
Resnik & Hart 2003 [28]	Outpatient rehab. clinics (USA)	OHS, PCS, PF-10	Patients treated in outpatient rehab. clinics for lower back pain	At intake and discharge from outpatient rehabilitation	Identify expert physical therapists	x	x	x	–	Yes, bivariate analysis to identify adjustors	Not reported	Generalized linear model with >7 patients per therapist required, then calculating the difference: observed—expected; no minimum number of patients per clinic required	Percentiles of differences reported
Sivaganesan et al., 2018 [24]	Sites performing spine surgery (USA)	ODI, EQ-5D, VAS-BP/VAS-LP	Patients treated with spine surgery	Before and 1 year after surgery	Inter-provider comparison for performance measuring	x	x	x	x	Yes, but no association coefficients are reported	Not reported	Random effects regression models and multilevel hierarchical Bayesian models (3 levels); no minimum number of patients per hospital required	Box plots for adjusted ranks per surgeon per site reported
Varagunam et al., 2015 [30]	NHS-funded hospitals (UK)	OHS, OKS, EQ-5D	Patients treated by consultant in NHS providers	Before and three or six months after surgery	Inter-consultant comparison for performance measuring	x	x	x	-	Not assessed	Not reported	Multiple linear regression, >39 patients per NHS provider and >9 patients per consultant required; multilevel models	Number of consultants reported that perform better than expected, as expected, and worse than expected

EQ-5D, European Quality of Life 5 Dimensions; IRES, *Indikatoren des Reha-Status* (Indicators for Rehabilitation Status); MDQ, Modified Low Back Pain Disability Questionnaire; MID, Minimally Important Difference; NDI, Neck Disability Index; NHS, National Health Service (UK); ODI, Oswestry Disability Index; OHS, Oxford Hip Score; OKS, Oxford Knee Score; PCS, Physical Component Summary; PF-10, Physical Functioning Subscale (of Short Form-36 questionnaire); SMFA, Short Musculoskeletal Function Assessment; VAS-BP, Visual Analog Scale for Back Pain; VAS-LP, Visual Analog Scale for Leg Pain. ^1^ According to Iezzoni, an eligible adjustor should 1. be associated with the outcome and 2. vary across providers [16].

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
