# Peer review of "Different Approaches for Case-Mix Adjustment of Patient-Reported Outcomes to Compare Healthcare Providers—Methodological Results of a Systematic Review"

_cancers, 2021, doi:10.3390/cancers13163964_

Round 1

Reviewer 1 Report

Thank you for the opportunity to review this article. It is an interesting study. Below please see my specific comments.

  1. Articles were included if they compared (a) different health-care providers using (b) case mix–adjusted (c) patient-reported outcomes. Please make clear if these criteria are AND or OR conditions.
  2. This sentence on page 2 last paragraph is not clear: “Secondly, while traditional quality metrics data are typically collected without — or with little — active contribution from the patient and are thus almost completely documented (although with varying documentation quality),…” Please revise.
  3. If “There have been few publications on how to adjust the case-mix for PROs”, please justify why it is important to do the review now that could provide useful information for future studies, especially if “Recommendations on the choice of future CMA methodology are also derived from this.” (Page 3, 2nd paragraph) As was stated in the discussion, there was lack of validation studies to base the recommendations on. 
  4. This exclusion criterion is not clear: those that used patient-reported experience measures (including satisfaction surveys) as opposed to the above definition of PROs. Please further explain what this exclusion criterion would entail. An example would be helpful that illustrate a PRO that does not fit the definition used in this article apart from satisfaction surveys.
  5. In Figure 1, for the 597 articles excluded, please add the number of references excluded under each individual criterion.
  6. Only MEDLINE was used in the search. Using just one database may miss potentially relevant articles. Please provide rationale for using only one database and whether missing potentially relevant articles is likely to be an issue for this study. For instance, the authors were surprised there were no cancer studies found. Was it because the selection criteria too restrictive or would there be any other databases that may have the relevant articles? 
  7. In the last column of Table 1, the summary table, some had R2 value but no information on statistical approach. What regression models were used to generate the R2? Without this information, it is hard to make sense of the R2. Also, to be helpful to readers who may not be very familiar with different PROC tools, a brief descriptions such the range of values from the score measured using different PROC tools would be helpful.
  8. Since examine the adjustors used is an important item for review, in addition to the number of variables as adjustors, it would be helpful to add the list of variables used as adjustors in Table 1 so that readers can see and compare what are the commonly used adjustors in the literature. Or, the authors could add that information in Table 2. Instead of “x” for which adjustors, the authors may include the variable names under each column.
  9. In Table 2, the column title of “how are adjustors chosen” is not consistent with the two sub-columns under it: is there an effect? is there inter-provider variation? This makes it harder to understand the information reported in those columns. For instance, for Sivaganesa’s study (25), under the column, “is there an effect”? The input is “x, but not reported”. Does that mean that there was an effect of the adjustors but the effects were not reported? Or, does it mean that a selection approach was used, but that approach was not reported, since this information was under the overall title of “How are adjustors chosen”? Some other entries in the this column included back-step regression approach as the approach to select adjustors. Please revise the titles of those columns.
  10. The paragraph above 3.8.3 is a not very clear. An illustration using the numbers similar to the illustrations provided for other approaches would be helpful.
  11. The first sentence on page 9, “Center C, which performed 10 points less well than expected, would be top-ranked following this approach”. Is that right? If only looking at the adjusted score, the interpretation would be the opposite of the other approaches. Is that right? If this is correct, then the authors needs to discuss this in comparison to other approach and comment on whether this approach is valid since it would indicate the opposite ranking to that using our approaches.

Author Response

Point-by-point response to reviewers’ comments

The authors gratefully thank the reviewers for their helpful comments. Please find below point-by-point responses in italic.

Reviewer 1

Thank you for the opportunity to review this article. It is an interesting study. Below please see my specific comments.

  • Articles were included if they compared (a) different health-care providers using (b) case mix–adjusted (c) patient-reported outcomes. Please make clear if these criteria are AND or OR conditions.
    • Thank you, we clarified in the abstract that a, b, and c were AND conditions.
  • This sentence on page 2 last paragraph is not clear: “Secondly, while traditional quality metrics data are typically collected without — or with little — active contribution from the patient and are thus almost completely documented (although with varying documentation quality),…” Please revise.
    • We transmitted this paragraph together with the previous one to the discussion section and rephrased some parts of it.
  • If “There have been few publications on how to adjust the case-mix for PROs”, please justify why it is important to do the review now that could provide useful information for future studies, especially if “Recommendations on the choice of future CMA methodology are also derived from this.” (Page 3, 2nd paragraph) As was stated in the discussion, there was lack of validation studies to base the recommendations on. 
    • Thank you for the opportunity to point out the relevance of this review: Since there are still few publications on how to perform CMA, we collected different approaches on how CMA is performed in the existing literature already. One main result of this review is the heterogeneity of CMA approaches and thus, we recommend creating a consensus-based methodological guideline; our recommendations could serve as suggestions for such a guideline. We specified this in the introduction.
  • This exclusion criterion is not clear: those that used patient-reported experience measures (including satisfaction surveys) as opposed to the above definition of PROs. Please further explain what this exclusion criterion would entail. An example would be helpful that illustrate a PRO that does not fit the definition used in this article apart from satisfaction surveys.
    • We included a footnote with the OECD definition of patient-reported experience measurements – in contrast to patient-reported outcomes – to clarify, why we only included studies using PROs.
  • In Figure 1, for the 597 articles excluded, please add the number of references excluded under each individual criterion.
    • Thank you for your valuable suggestion; we included those information in the figure.
  • Only MEDLINE was used in the search. Using just one database may miss potentially relevant articles. Please provide rationale for using only one database and whether missing potentially relevant articles is likely to be an issue for this study. For instance, the authors were surprised there were no cancer studies found. Was it because the selection criteria too restrictive or would there be any other databases that may have the relevant articles? 
    • MEDLINE is the most applicable database for the purposes of this article. We additionally screened references of the included articles to account for any articles which were not found by our MEDLINE search. However, we discussed this issue in the limitation section.
  • In the last column of Table 1, the summary table, some had R2 value but no information on statistical approach. What regression models were used to generate the R2? Without this information, it is hard to make sense of the R2. Also, to be helpful to readers who may not be very familiar with different PROC tools, a brief descriptions such the range of values from the score measured using different PROC tools would be helpful.
    • Information on type of model can be found in table 2. For readability reasons, we do not include those information additionally in table 1.
      In this review, more than 15 different PRO instruments were included. Most of them have different ranges of values. Including those information could be more confusing for readers, that are merely interested in the statistical approaches of CMA than in the actual PRO tools used. To interpret R2 and the different statistical approaches, the specific PRO ranges are not of importance (e.g., R2 is independent of the range of the dependent variable).  
  • Since examine the adjustors used is an important item for review, in addition to the number of variables as adjustors, it would be helpful to add the list of variables used as adjustors in Table 1 so that readers can see and compare what are the commonly used adjustors in the literature. Or, the authors could add that information in Table 2. Instead of “x” for which adjustors, the authors may include the variable names under each column.
    • Thank you for your valuable suggestion. We included information on the specific adjustors in table 1.
  • In Table 2, the column title of “how are adjustors chosen” is not consistent with the two sub-columns under it: is there an effect? is there inter-provider variation? This makes it harder to understand the information reported in those columns. For instance, for Sivaganesa’s study (25), under the column, “is there an effect”? The input is “x, but not reported”. Does that mean that there was an effect of the adjustors but the effects were not reported? Or, does it mean that a selection approach was used, but that approach was not reported, since this information was under the overall title of “How are adjustors chosen”? Some other entries in the this column included back-step regression approach as the approach to select adjustors. Please revise the titles of those columns.
    • Thank you for the opportunity to revise table 2: According to Iezzoni, an eleigibale adjustor should 1.  be associated with the outcome and 2. vary across  providers. We therefore re-named the column (“Are criteria for the selection of adjustors met?” with a footnote explaining that the criteria used are according to Iezzoni, and sub-titles: Are the adjustors associated with the outcome?“ and “Do adjustors vary across providers?”).
  • The paragraph above 3.8.3 is a not very clear. An illustration using the numbers similar to the illustrations provided for other approaches would be helpful.
    • Thank you for your suggestion; we included an example.
  • The first sentence on page 9, “Center C, which performed 10 points less well than expected, would be top-ranked following this approach”. Is that right? If only looking at the adjusted score, the interpretation would be the opposite of the other approaches. Is that right? If this is correct, then the authors needs to discuss this in comparison to other approach and comment on whether this approach is valid since it would indicate the opposite ranking to that using our approaches.
    • Yes, your interpretation is right! We totally agree with your interpretation of this result and pointed this problem out in the discussion section: “One study only calculated fitted values and based the resulting provider ranking on them, which seems to be a questionable adjustment approach since providers with a “favorable” casemix (e.g., younger or healthier patients) will always perform better than providers with “difficult” casemix if using this CMA approach. This can result in a biased ranking and does not provide information on the actual quality of care.

Reviewer 2 Report

The manuscript was prepared very well. The introduction section justifies the purpose of the study. I congratulate the authors for the preparation of the manuscript.

However, I have the following comments:

Introduction

  • What do the studies (5,6,7) imply for the development of new approaches to incorporate (electronic) PROs into routine clinical cancer care? and what do these new approaches to incorporate (electronic) PROs into routine clinical cancer care consist of? and do they differ from those of 15 years ago? explain this question.
  • What is meant by quality assurance? include some reference in the first lines of the first paragraph on page 23.
  • CMA of PRO comparisons has to deal with two additional major challenges:

Firstly and Secondly, - include some reference.

  • The introduction is too long and with contents that could be used in the discussion. Please summarize it and focus on aspects related to the objective of the manuscript.

Methodology

  • Please include the search string with the month terms used.
  • Include a table with the number of:

Search number          

Databases used          

Search terms 

Number of articles    

Number of articles after application of filters in the databases Number of articles selected after application of inclusion and exclusion criteria

  • Why do I use only Medline?

Results

  • The number of articles included is 9, why do you add 2 studies by another way?
  • In the abstract you indicate that there are 12 articles included and in the results 11, please clarify this issue.
  • You should perform a methodological quality assessment by another 7 other methods to give robustness to your manuscript. You can use, PEDRO, Mc MASTER....
  • Please improve the content of table 1. There is information that is difficult to understand. Why do some studies have no information in the columns? please review the abbreviations in table 1 and include all of them.

Discussion

In the first part you should add something about the purpose of the review and the main results. You should also differentiate between the limitations (a separate section) and the results.

What is the application of your study? include it.

Include references in those paragraphs that are not your own ideas or discussions.

Author Response

Point-by-point response to reviewers’ comments

The authors gratefully thank the reviewers for their helpful comments. Please find below point-by-point responses in italic.

Reviewer 2

The manuscript was prepared very well. The introduction section justifies the purpose of the study. I congratulate the authors for the preparation of the manuscript.

However, I have the following comments:

Introduction

  • What do the studies (5,6,7) imply for the development of new approaches to incorporate (electronic) PROs into routine clinical cancer care? and what do these new approaches to incorporate (electronic) PROs into routine clinical cancer care consist of? and do they differ from those of 15 years ago? explain this question.
    • Thank you for this question. The purpose of this review is not to examine the possibilities of PROs for the patient-individual routine cancer care and the opportunities of electronic PROs; that is the reason why we do not discuss those in detail in our article. We included the mentioned references as a theoretical framework on how PROs are already proposed to be used in clinical care.
  • What is meant by quality assurance? include some reference in the first lines of the first paragraph on page 23.
    • Accoarding to the OECD, quality assurance is: “A planned and systematic pattern of all the actions necessary to provide adequate confidence that a product will conform to established requirements.” (https://stats.oecd.org/glossary/detail.asp?ID=4954). To clarify our statement, we included “in the sense of quality improvement initiatives”.
  • CMA of PRO comparisons has to deal with two additional major challenges:

Firstly and Secondly, - include some reference.

  • We transferred this paragraph to the discussion section, as you proposed below, and pointed out that those “challenges” were based on the authors’ opinion.
  • The introduction is too long and with contents that could be used in the discussion. Please summarize it and focus on aspects related to the objective of the manuscript.
    • Thank you for this valuable suggestion. As written above, we transferred some parts of the introduction to the discussion and conclusion section.

Methodology

  • Please include the search string with the month terms used.
    • We did not use any date restriction, as this was not applicable for the purposes of this review. The search string is included in Appendix 1.
  • Include a table with the number of:

Search number          

Databases used          

Search terms 

Number of articles    

Number of articles after application of filters in the databases Number of articles selected after application of inclusion and exclusion criteria

  • We included information on the number of hits for the three parts of the search string in the appendix. The other information are part of the revised figure 1 and the methods section (2. 3 search and study selection).
  • Why do I use only Medline?
    • MEDLINE is the most applicable database for the purposes of this article. We additionally screened references of the included articles to account for any articles which were not found by our MEDLINE search. However, we discussed this issue in the limitation section.

Results

  • The number of articles included is 9, why do you add 2 studies by another way?
    • As proposed by Cochrane, we screened the references of the included studies (https://www.cochrane.org/MR000026/METHOD_examining-reference-lists-to-find-relevant-studies-for-systematic-reviews). 
  • In the abstract you indicate that there are 12 articles included and in the results 11, please clarify this issue.
    • Thank you for finding this typo! We corrected the number.
  • You should perform a methodological quality assessment by another 7 other methods to give robustness to your manuscript. You can use, PEDRO, Mc MASTER....
    • Thank you for your suggestion. Most quality assessment tools, as PEDRO for instance, are designed for RCTs and as such not applicable for the kind of studies we included (retrospective analysis of prospectively collected data, mostly in the sense of observational cohorts). We thus applied the CASP tool to assess study quality, as described in the article. Other tools designed for cohort studies, for instance the NOS scale, propose numerical rating, but include some items, which are not applicable to our studies either (e. g., assessment of exposure). We discussed this issues in the limitation section for clarification.
  • Please improve the content of table 1. There is information that is difficult to understand. Why do some studies have no information in the columns? please review the abbreviations in table 1 and include all of them.
    • We included information in all columns, if evaluable. We revised abbreviations and included more information on the specific adjustors.

Discussion

In the first part you should add something about the purpose of the review and the main results. You should also differentiate between the limitations (a separate section) and the results.

  • We separated the discussion section in three parts: discussion, limitations and conclusions/recommendations and specified the first sentence to clarify the purpose of the review.

What is the application of your study? include it.

  • The purpose of this systematic review is the collection and synthesis of existing CMA approaches. As a consequence of our findings, we suggest developing a consensus-based guideline, as proposed in the Conclusions and Recommendations Section. Furthermore, we included recommendations on further research in this section. For stakeholders looking for a possible CMA approach, our findings describe the already existing and peer-reviewed published ones.

Include references in those paragraphs that are not your own ideas or discussions.

  • We clarified the section which are based on the authors’ ideas (mostly in the conclusion and recommendation section). Other literature-based statements are referenced.

Round 2

Reviewer 2 Report

 there are no more suggestions